palaeontology, ecology

biogeography, palaeobiogeography, conservation palaeobiology, extinction, beta diversity, species richness

**Author for correspondence:**
Simon A. F. Darroch
e-mail: simon.a.darroch@vanderbilt.edu

### PUBLISHING

# The preservation potential of terrestrial biogeographic patterns

Simon A. F. Darroch[1,2], Danielle Fraser[3,4,5,6] and Michelle M. Casey[7]

[1]Department of Earth and Environmental Sciences, Vanderbilt University, 5726 Stevenson Center, Nashville, TN 37240, USA
[2]Senckenberg Museum of Natural History, Frankfurt 60325, Germany
[3]Department of Palaeobiology, Canadian Museum of Nature, 240 McLeod Street, Ottawa, Ontario, Canada K2P 2R1
[4]Department of Biology, and [5]Department of Earth Sciences, Carleton University, 1125 Colonel By Drive, Ottawa, Ontario, Canada K1S 5B6
[6]Department of Paleobiology, National Museum of Natural History, Smithsonian Institution, 10th and Constitution NW, Washington, DC 20560-0121, USA
[7]Department of Physics, Astronomy and Geosciences, Towson University, 8000 York Road, Towson, MD 21252, USA

SAFD, 0000-0003-1922-7136

Extinction events in the geological past are similar to the present-day biodiversity crisis in that they have a pronounced biogeography, producing dramatic changes in the spatial distributions of species. Reconstructing palaeobiogeographic patterns from fossils therefore allows us to examine the long-term processes governing the formation of regional biotas, and potentially helps build spatially explicit models for future biodiversity loss. However, the extent to which biogeographic patterns can be preserved in the fossil record is not well understood. Here, we perform a suite of simulations based on the present-day distribution of North American mammals, aimed at quantifying the preservation potential of beta diversity and spatial richness patterns over extinction events of varying intensities, and after applying a stepped series of taphonomic filters. We show that taphonomic biases related to body size are the biggest barrier to reconstructing biogeographic patterns over extinction events, but that these may be compensated for by both the small mammal record preserved in bird castings, as well as range expansion in surviving species. Overall, our results suggest that the preservation potential of biogeographic patterns is surprisingly high, and thus that the fossil record represents an invaluable dataset recording the changing spatial distribution of biota over key intervals in Earth History.

## 1. Introduction

Extinction events in the geological past are thought to have had a pronounced biogeography, including spatial heterogeneity in rates of extinction and recovery [1–5], changes in the geographical range sizes of affected species [6–8], and changes in beta diversity on a wide range of spatial scales [9–12]. Similarly, the anthropogenic factors contributing to the present-day biodiversity crisis (which has also been called the '6th mass extinction' [13]) are biogeographically complex [14,15], and are having global-scale effects on the spatial distributions of species [16–19]. Given that many contemporary proximal extinction drivers are also recognized as having driven biotic crises in the past [20], reconstructing biogeographic patterns over past extinction events may represent a powerful tool for building predictive and spatially explicit models of biodiversity loss [2], allowing us to address the questions: where will extinction be most severe? How will biogeographic patterns be altered? And, which regions are in the most urgent need of protection? These questions comprise a major research area within the field of 'conservation palaeobiogeography' [21], and are an area where palaeontological data can be used to understand the effects of ongoing global change.

The extent to which biogeographic patterns can be preserved in the fossil record, however, has only just begun to be investigated [22–25]. A wide variety of factors, including the changing spatial and temporal distributions of fossiliferous sediment, the distribution of fossil localities, and differential taphonomic potentials of species may all exert controls over how accurately spatial diversity patterns can be preserved (e.g. [26]). Here, we examine whether two central facets of biogeography—beta diversity and spatial richness patterns—can be reliably preserved in fossil locality data over a range of hypothetical extinction scenarios, and when accounting for a number of taphonomic processes.

Beta diversity was originally conceived to describe variation in taxonomic composition across space [27] and is central to addressing the processes underlying the formation of local and regional biotas [28–32]. In the geological past, beta diversity has been shown to change in response to extinction, origination, immigration, and shifts in the geographical ranges of species [9,11,12]. In modern settings, changes in beta diversity have been shown to be a sensitive indicator of ecological stress [33], and, consequently, studies using beta diversity underpin much of conservation theory and practice [29,32]. Understanding how beta diversity changed over past biotic crises may therefore help to identify modern areas undergoing ecological stress, or ecosystems at risk of incipient collapse [11,33,34].

Spatial richness patterns, in contrast, represent a different facet of biogeography, and a separate set of conservation priorities. The fact that species are not distributed homogeneously in space is a fundamental observation in ecology, with huge efforts dedicated to determining the biotic, abiotic, and historical controls on richness patterns at a broad range of spatial scales [35]. Areas of high local richness (which may or may not strictly be biodiversity 'hotspots', see e.g. [36]) necessarily represent conservation priorities [37–39]; being able to reliably identify such areas in the past would thus allow palaeontologists to establish whether highly biodiverse areas are prone to moving, expanding, or shrinking in a variety of global change scenarios, or are more (or less) vulnerable to mass extinction (e.g. [40]).

This study, therefore, represents a first attempt to calibrate the extent to which we can use the terrestrial fossil record to predict the biogeographic effects of ongoing global change, both in terms of the effects of extinction on beta diversity and changes to spatial richness patterns. We employ a simulation-based framework based on known patterns of extant North American mammal distributions and the distribution of fossil localities from a key interval of the Cenozoic. We apply a stepped series of preservation biases based on published relationships between mammal body mass, population density, and expected number of carcasses [41–43], and simulate extinction of varying intensities (25%, 50%, and 75% species extinction) by removing small-ranged taxa. We then calculate metrics for beta diversity and map patterns of richness as tests of how well we can detect known biogeographic patterns using a biased fossil record.

## 2. Material and methods

### (a) Species

In order to preserve realistic range geometries and size distributions in simulations, we use the polygon distributional data

for 374 extant terrestrial mammal species (taken from the International Union for Conservation of Nature (IUCN) Redlist: www.iucnredlist.org/) whose ranges extend into North America (i.e. the USA, Canada, and Mexico); these species describe an approximately lognormal distribution of range sizes (electronic supplementary material, S1), which is typical for a majority of terrestrial taxonomic groups [44–47].

### (b) Simulating reductions in beta diversity

We first rasterized each species' range using a 1° latitude and longitude grid, and then summed diversity within grid cells across North America; this process produced a species richness map for our mammal species that served as the 'baseline' for experiments (figure 1). We calculated beta diversity for this baseline as a function of multisite Sørensen's dissimilarity using species lists for every occupied grid cell. Multisite metrics (such as Sørensen's dissimilarity) are similar to Whittaker's [27] original formulation, and account for compositional heterogeneity for assemblages of more than two sites [48,49].

In this study, we assume extinction as a driver of change in beta diversity. Although extinction is not the only regional-scale process operating on geological (rather than ecological) timescales that can affect beta diversity—origination [50], climate [10], faunal immigration [8], and tectonism [51,52] have all been shown to have effects on the differentiation of faunas in the geological past— extinction events are overwhelmingly thought to preferentially remove small-ranged species [6,12,53–55], reducing endemism and thus faunal heterogeneity. To simulate extinction, we sequentially remove 25%, 50%, and 75% of the smallest ranged species in our baseline distribution; this results in three different extinction scenarios with an increasing preponderance of large-ranged species, and more similar species pools across local assemblages.

### (c) Beta diversity experiments

We test whether extinction-driven reductions in beta diversity can be reliably preserved in the fossil record under five different experimental conditions, each of which adds a layer of complexity and realism in simulations, and thus allows us to identify the source(s) of uncertainty in reconstructing biogeographic patterns.

In our first experiment (1, 'random localities'), we iteratively (x100) and randomly place simulated fossil sites within North America (see also [23,24]). We then extract species from these sites in each iteration, and calculate beta diversity in the same fashion as in our baseline. To test whether results are sensitive to the number of fossil 'localities', we run simulations using 3, 30, and 300 sites; these numbers represent the upper and lower range of the actual number of mammal fossil localities typically recorded in Cenozoic time slices, and which have been used in prior palaeobiogeographic analyses [8,56]. Repeating this exercise using our three extinction scenarios allows us to test whether resulting changes in beta diversity can be reliably recorded in simulated fossil localities.

In our second experiment (2, 'real localities'), we account for the patchy distribution of fossil localities in time and space [26] by restricting fossil sites to the latitude and longitude coordinates of actual fossil localities from the Rancholabrean North American Land Mammal Age (Late Pleistocene—240–11 thousands of years ago (kya)). This interval was chosen because of the large number of known fossil localities (allowing us to employ our 3, 30, and 300 site sampling protocol; figure 1), and because it is potentially a critical interval for assessing drivers of the Pleistocene extinction of megafauna [57].

In our third experiment (3, 'taphonomy'), we add a series of equations modified from Fraser [22] to our simulated sites, that controls for differing taphonomic potentials among mammal species by estimating a 'probability of preservation'. The equations build on a number of macroecological correlates for

**Figure 1.** Flow chart illustrating the methods employed for reconstructing changes in the spatial organization of biota with increasing extinction intensity. (Left) Mammalian species richness is first mapped onto 1° grid cells for 0, 25%, 50%, and 75% extinction scenarios. (Centre) In each scenario 3, 30, and 300 'localities' are iteratively simulated; in our first experiments (random) these are placed randomly, while from our second experiment (localities) onwards, these are chosen from existing Rancholabrean fossil sites (shown here). Species lists compiled from simulated sites are then passed through a stepped series of taphonomic filters (taphonomy, lagerstätten, and castings). (Right) In each iteration, final species lists are then used to calculate beta diversity as a function of multisite Sørensen's index, and also used to map diversity back onto continental North America (here using a $3 \times 3$ grid moving window interpolate richness across empty grid cells). These simulated beta diversity and richness patterns are compared back to original extinction scenarios, to see how accurately biogeographic changes associated with extinction are recorded in simulations. (Online version in colour.)

population density [41,42], body mass, and death rate [43] to produce an estimate for the number of carcasses for each species:

$$\log D = -0.75 \log W + 4.23 \quad [43] \qquad (2.1)$$

$$d_r = 3.09 W^{-0.33} \quad [41] \qquad (2.2)$$

$$\text{and} \quad C_e = d_r 10^{\log D} \quad [42], \qquad (2.3)$$

where $W$ is body mass (g), $D$ is the population density, $d_r$ is the death rate, and $C_e$ is the expected number of carcasses for a given species. Probability of preservation was then summarized as the ratio of sampled to expected carcasses:

$$\log \frac{F_{s'}}{F_e} = -1.720 + 0.683 \log W \quad [42], $$

where $W$ is body mass (in kg) and $F_{s'}/F_e$ is the ratio of sampled to expected carcasses.

This series of equations therefore accounts for body size and relative rarity, both of which exert a strong control on the likelihood of fossil preservation in different species, and thus the reliability of reconstructed biogeographic patterns [22]. Applying the above equations to the 374 species used in this analysis produces an approximately lognormal distribution of preservation potentials, with the vast majority of species exhibiting low chances of fossilization (electronic supplementary material, S2).

We use the above series of equations in simulations by establishing a random sampling vector between 0 and 1 divided into 0.01 increments; for every instance where a simulated fossil 'site' intersects with a species range, we count that species as 'found' at that site if a random number taken from our sampling vector is equal to or lower than that species' probability of preservation.

All experiments thus far assume similar taphonomic potentials across all localities, and do not account for potential fossil lagerstätten, which might preserve all species equally regardless of body mass and/or rarity. In our fourth experiment (4, 'lagerstätten'), we therefore address the conservative nature of our equation for 'preservation potential' by additionally simulating the presence of rare fossil lagerstätten (which in the Rancholabrean could reasonably represent cave deposits—see e.g. [58]). To do this, we include another function which introduces a 1% probability that any simulated fossil locality is characterized by exceptional preservation, and thus preserves every species that occurs there. This 1% figure reflects a conservative estimate for the rarity of fossil lagerstätten [59].

Lastly, our equation for calculating differing taphonomic potentials among species represents a significant bias against the preservation of taxa with small body sizes [60–62]. However, in addition to more conventional fossil lagerstätten (such as the

Rancho La Brea tar pits in Los Angeles county), which are simulated in our fourth experiment, the pellets regurgitated by owls and other birds (castings) are an important geological deposit that overwhelmingly preserve small animals, and provide valuable insights into small mammal communities [63–66]. In our fifth experiment (5, 'castings'), therefore, we simulate the collection of bird castings from all localities by recording all species between 5 and 800 g body mass (which represents a typical range of prey size for medium-sized owls [67]) as 'found'.

We illustrate simulated beta diversity as boxplots, where boxes contain the first and third quartiles of simulated diversity estimates, whiskers illustrate the maximum and minimum, and notches give 95% confidence intervals around the median. We define a change in beta diversity in any of our extinction scenarios as 'likely to be detected' if the boxes on boxplots show no overlap with our baseline scenario (i.e. less than 25% overlap in the total range of diversity values), and thus a high probability that change in beta will be preserved in the fossil record.

## (d) Reconstructing spatial richness patterns

Finally, we test whether continental-scale biogeographic patterns in our baseline maps can be accurately reproduced in our five experimental scenarios, by extracting species lists from simulated fossil localities and reconstructing diversity across 1° grid cells. We calculate the agreement between 'true' and simulated richness rasters in each iteration using a rank-based and non-parametric correlation test—Kendall's Tau—producing a vector of agreement estimates, whereby higher Tau values indicate greater similarity between 'true' and simulated diversity rasters. We calculate agreement first in a simple grid-wise fashion (Simulated), and also using a smoothing function on our simulated richness raster to interpolate richness values across empty cells (Extrapolated); this function employs a 3 × 3 grid cell moving window, then calculates the mean richness for each 9-cell block and assigns this value to the central cell.

## (e) Additional tests

We subject the results of our 'castings' beta diversity experiments to two additional tests that examine the preservational fidelity of changes in beta diversity to different patterns of extinction selectivity and/or biotic response.

First, extinction events are typically thought to select against small-ranged taxa (leaving more species common to all local species pools and thus reducing beta diversity). However, surviving species, or new species evolving in the aftermath, can experience an increase in range size as they proliferate and disperse in response to the availability of free ecospace [68,69]. Range expansion among surviving taxa would add more species common to all local assemblages, and thus exacerbate reductions in beta diversity. We test the strength of this effect by expanding the ranges of the top 5%, 10%, and 20% of the largest-ranged survivors (for 25%, 50%, and 75% extinction thresholds, respectively) to fill that of North America, thus simulating dramatic range expansion among surviving species, and/or replacement by extremely cosmopolitan and invasive taxa.

Second, a key question is: given the space- and taphonomy-related biases inherent in fossil preservation, to what extent might random extinction of species lead to changes in beta diversity, and to what extent might these changes be preserved in the fossil record? To test this, we simulate 'random' extinction whereby 25%, 50%, and 75% of species are removed with no selectively for range size. Note that, given that species are removed at random, this test necessitated iteratively (×100) removing random species to generate baseline trends, in order to explore the distribution of possible results.

All analyses were performed in the statistical software environment R [70].

# 3. Results

## (a) Baseline beta and biogeographic patterns

Our three extinction scenarios (25%, 50%, and 75% extinction) result in a decline in beta diversity. This decline is relatively minor in our 25% and 50% extinction experiments, but more severe when 75% of the smallest ranged species are removed (figure 2a). In terms of diversity patterns, our baseline map for extant mammal species richness shows a clear area of high richness in the southwest USA and northern Mexico, and a broad gradient characterized by high richness in the west (roughly corresponding to the eastern extent of the Rocky Mountains), and low richness in the east and southeast (figure 1). This broad east-west gradient in richness is still apparent at 75% 'extinction', illustrating that this pattern is influenced by large-ranged species, as well as small-ranged species.

## (b) Beta diversity experiments

Our ability to detect reductions in beta diversity associated with extinction improved with more sampled fossil localities.

Using the multisite Sørensen index (figure 2), reductions in beta diversity associated with 50% and 75% extinction were readily detected (i.e. boxes display less than 25% overlap) using 30 localities in 'random locality' experiments, while only 75% extinction could be detected in 'real localities'. No reduction in beta diversity was detectable with the addition of either 'taphonomy' or 'lagerstätten'. In our last experiment (5, 'castings') a reduction in beta was once again detectable at 75% extinction. Using 300 localities, reductions in beta diversity associated with 50% and 75% extinction can be detected in both experiments 1, 'random localities' and 2, 'real localities'. With the addition of our first two taphonomic filters (3, 'taphonomy' and 4, 'lagerstätten') only 75% extinction could be detected, while in our final experiment (5, 'castings') both 50% and 75% extinction thresholds were detectable.

No reductions in beta diversity were detected using only three fossil localities, nor was the 25% extinction threshold detected in any experiments.

## (c) Reconstructing species richness patterns

Kendall's Tau correlations between simulated and 'true' richness patterns (figure 3) illustrate high correlation ($Tau = 0.7$–$0.9$) in our first and second experiments (1, 'random localities' and 2, 'real localities'), but much lower correlations ($Tau = 0.0$–$0.4$) in our third experiment (3, 'taphonomy'). With the addition of lagerstätten, Tau values increase slightly, while remaining lower than the first two experiments. Lastly, in our fifth experiment (5, 'castings') correlations once again become higher ($Tau = 0.4$–$0.8$). In general, there is little tendency for increase/decrease in correlation with increasing extinction intensities, with the exception of 'castings', where correlations between simulated and 'true' richness patterns are lower ($Tau = {\sim}0.45$ as opposed to $Tau = {\sim}0.55$) at our 75% extinction threshold than other scenarios.

## (d) Additional tests

The results of additional tests are shown in figure 4. Our first additional test—simulating range expansion in surviving taxa—produces a baseline scenario where decrease in beta

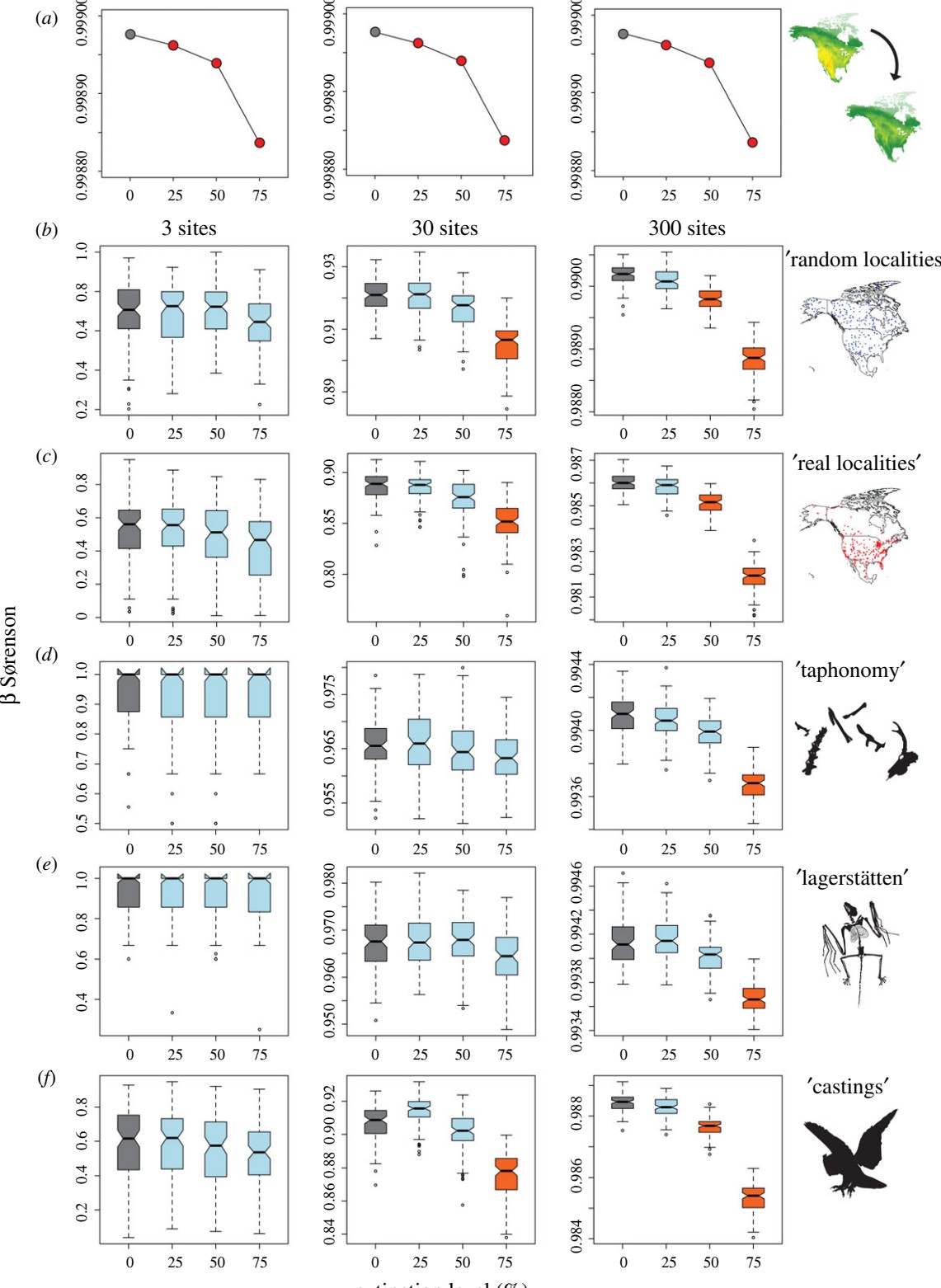

**Figure 2.** 'True' and simulated changes in beta diversity with simulated extinction. (*a*) 'True' decreases in beta diversity produced by simulated extinction, measured as multisite Sørensen's dissimilarity. 0% extinction marked in grey to indicate that this is the 'baseline' scenario. (*b–f*) Simulated beta diversity for our 5 experiments (random, real localities, taphonomy, lagerstätten, and castings) illustrated as boxplots; boxes give the interquartile range (IQR), while whiskers give 1.5x IQR, and outliers plotted as points. Baseline (i.e. 0% extinction) scenario illustrated in grey. Boxplots coloured blue indicate those that overlap with the baseline and thus are unlikely to be detectable in the fossil record. Boxes coloured orange possess no overlap with the baseline, and thus indicate scenarios where change in beta diversity is most likely detectable.

diversity with increasing extinction intensity is more linear than in our original experiments (for example, contrast figure 2*a* with figure 4*a*). Our 'castings' experiments under these conditions readily detect these reductions in beta diversity, recovering significant decreases at all extinction intensities. Our second additional test—simulating random, rather than range-selective extinction—produces a very different baseline scenario, with no obvious trend in beta diversity with increasing extinction intensity (save increasing variance, reflecting the differing distributions of large- versus

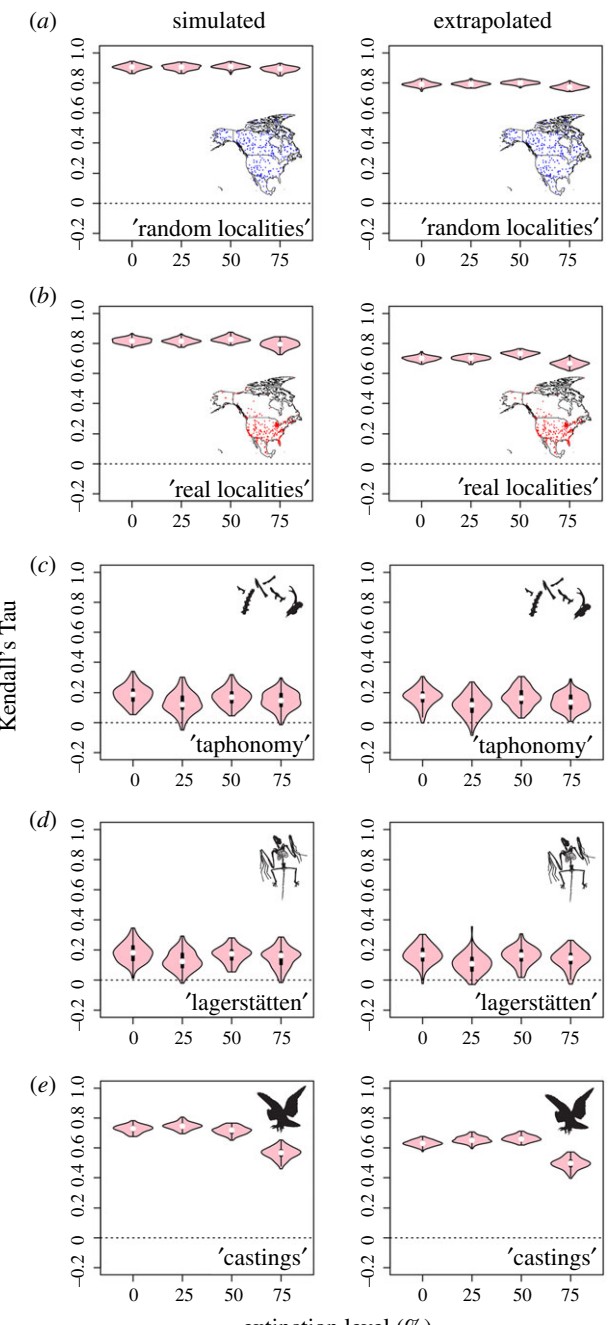

**Figure 3.** Spatial correlations (as Kendall's Tau) between 'true' and simulated richness maps. Tau values illustrated as violin plots, both grid-wise (Simulated - left) and (right) 'Extrapolated' where a 3 × 3 grid cell moving window approach has been employed. Higher Tau values indicate better agreement between 'true' and simulated richness maps. Results of experiments 'random', 'real localities', 'taphonomy', 'lagerstätten', and 'castings' given in rows a–e, respectively. Violin plots illustrate the distribution of correlations after 100 iterations in each experiment. (Online version in colour.)

small-ranged taxa removed in each iteration). Repeating our 'castings' experiment under these conditions reveals a close match with the baseline results (i.e. no consistent trend).

## 4. Discussion

At first glance, our results would suggest that the preservation potential of biogeographic patterns, both in terms of identifying decreases in beta diversity associated with extinction and the location of biogeographic richness patterns, are

relatively low. Plotting estimates of gamma diversity (i.e. total North American diversity) over the same set of simulations (electronic supplementary material, S3) shows that, even when an extinction signal is captured in gamma (i.e. the overall number of recovered species), a decrease in beta diversity is not always detected. Even in our most optimistically modelled scenario (i.e. 300 localities, where bird castings are recovered from every simulated locality), we can only reliably identify decreases in beta diversity associated with 50% and 75% extinction, representing severe biotic crises (potentially 'mass extinctions'). In a similar vein, our diversity reconstruction experiments recover Kendall's Tau values of approximately 0.6, which, although significant ($p$-values consistently less than 0.05), indicate an imperfect replication of richness patterns across North America. However, for several reasons, we argue that these results are conservative, and that the potential for recovering deep-time changes in biogeographic patterns is higher than these numbers would suggest.

For example, we note that the actual decrease in beta diversity associated with lower extinction intensities is relatively small; 50% species extinction, for example, only results in an approximately 5% decrease in regional beta diversity, which is naturally hard to detect. In spite of this, our first additional test illustrates that a reduction in beta diversity associated with 25% species extinction can be detected on continental scales, when only 5% of surviving taxa undergo range expansion (figure 4a). Setting post-expansion ranges to fit the entirety of North America is arbitrary, but our 5% figure is certainly conservative (in the aftermath of late Pleistocene extinction approximately 70% of surviving species significantly expanded their ranges – see [71]). The results of our experiments thus illustrate that range expansion amongst a relatively small proportion of surviving taxa can significantly contribute to decrease in beta diversity, and this decrease likely be recorded in fossil locality data.

We acknowledge, however, that several aspects of our study are idealized. For example, in simulating extinction events of varying intensities, we remove the 25%, 50%, and 75% smallest ranged taxa. Although past biotic crises have shown a general tendency to preferentially select against small-ranged species (e.g. [54]), this tendency is rarely perfect, and over some extinction events selectivity was more or less random with respect to geographical range size [53,72,73]. Our second additional test provides some insight into the potential impact of this different selectivity scenario. Baseline results for this test illustrate that random extinction does not produce consistent increases or decreases in beta diversity (figure 4b). This is faithfully replicated in our simulation experiments, which likewise illustrate no consistent tendency for beta diversity to change with increasing extinction intensity. The large variance in simulated beta at the 75% extinction threshold does suggest that there may a slight risk of false positive signals at high extinction intensities (and when selectivity is random with respect to range size), but also that this is unlikely to be a pervasive issue through longer swathes of geological time. Lastly, although the vast majority of studies have identified small range size as a predictor of extinction risk, some studies have suggested stronger correlations with other macroecological attributes, such as body size [74,75]. Future studies should therefore further investigate extinction risk factors other than geographical range, including life-history traits [76]. In

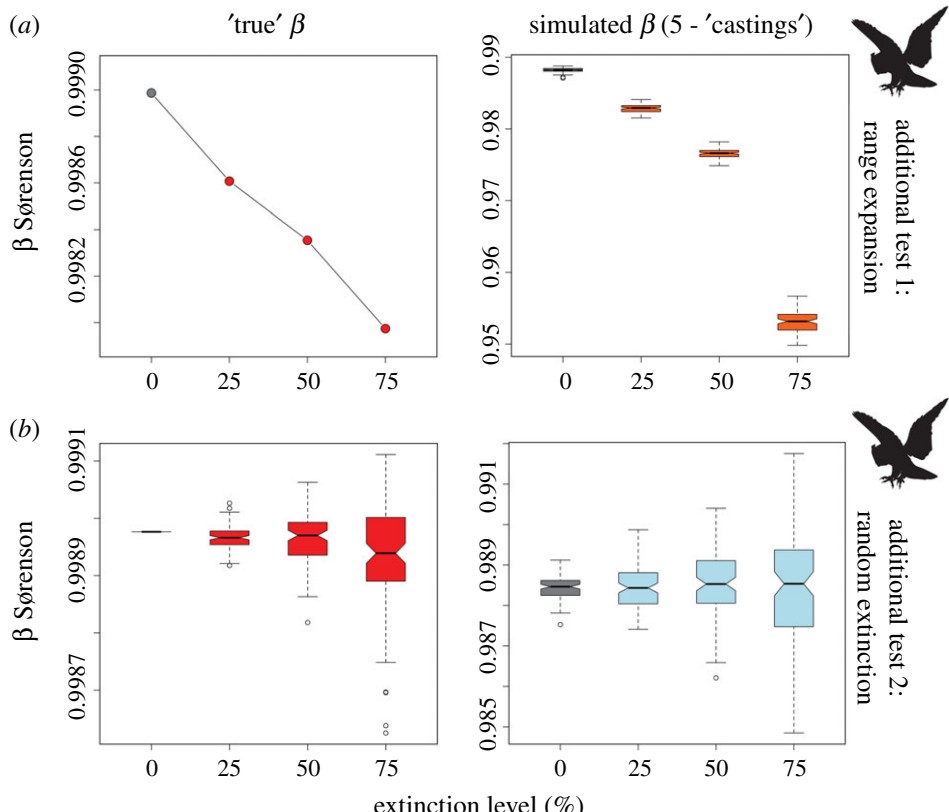

**Figure 4.** Simulated beta diversity for our two additional 'castings' experiments (performed using 300 sites); left column gives 'true' beta patterns with increasing extinction intensity (i.e. equivalent to plots shown in figure 2a), while the right column illustrates beta patterns generated in simulations. (a) Extinction of small-ranged species combined with range expansion of 5%, 10%, and 20% of the largest-ranged surviving species (for 25%, 50%, and 75% extinction thresholds, respectively). (b) Null scenario, where extinction is random with respect to range size. Baseline (i.e. 0% extinction) scenario illustrated in grey. Boxplots coloured blue indicate those that overlap with the baseline and thus are unlikely to be detectable in the fossil record; those coloured orange possess no overlap with the baseline, and thus indicate scenarios where change in beta diversity is most likely detectable. Boxes give the interquartile range (IQR), while whiskers give 1.5x IQR, and outliers plotted as points. (Online version in colour.)

addition, a preliminary investigation into the preservation potential of beta diversity patterns when large-ranged species are driven extinct reveals some interesting contradictions, although we consider this scenario unlikely in the real world (electronic supplementary material, S4–S5). We do, however, note that small range size has been shown to be a significant predictor of extinction risk in the current biodiversity crisis (e.g. [77]), and so argue that our simulations have at least some present-day relevance.

With measures of beta diversity, the biggest decrease in fidelity comes with inclusion of a taphonomic body size bias; taphonomic processes in terrestrial settings overwhelmingly bias against the preservation of small mammals [61], and the vast majority of the North American mammal fauna possess small body sizes [78–80]. Mechanisms for preserving the small mammal record are thus critical to preserving biogeographic patterns in deep time. In this regard, our choice of 1 in 100 localities representing lagerstätten is an approximation, but the actual number is potentially far lower (although, as isolated point sources of diversity lagerstätten are unlikely to be useful, as they would likely create apparent hotspots that may or may not be representative of regional biogeographic patterns). In addition, our best results are obtained through simulating the preservation (and recovery) of bird castings at every locality; however, bird castings are not a common feature of the fossil record [81–83], and, even when found, may not sample every small mammal within the local assemblage. Although bird castings

are well known from Pleistocene deposits, they are less well known from older time slices (the oldest described pellets are Oligocene [84,85]), potentially making this taphonomic window relevant only to the latter parts of the Cenozoic. However, others argue that this more likely represents a collection (and recognition) bias—gastric pellets have a number of properties that should favour fossilization, but may lack visual characteristics that would ordinarily lead to their collection [82]. In addition, other authors have noted that a wide variety of predatory vertebrates (in addition to birds), including varanid lizards, crocodilians, marine mammals, and potentially even several extinct groups all produce gastric pellets in a similar fashion [82,85,86], potentially creating a small vertebrate fossil record from a range of palaeoenvironments stretching back prior to the Oligocene. The fossil record of small mammals in castings may thus be much richer and more complete than is commonly thought, and raises the possibility that biogeographic patterns may be accurately reconstructed for key intervals in deep time.

In summary, extinction events in the geological past have had a pronounced biogeography, and understanding the spatial responses of biota to extinction in the geological past may have potential to help interpret current patterns of biodiversity loss [2]. Our study illustrates that: (a) the preservation potential of biogeographic patterns—i.e. changes in beta diversity and spatial richness patterns—over simulated moderate to intense extinction events (greater than 50% species loss) may be surprisingly high. Although decreases

in beta driven by low to moderate (25–50%) extinction of small-ranged species are relatively minor, range expansion in surviving 'disaster' taxa exacerbate this effect, and these reductions can be detected under our most optimistic scenarios. In addition, the biggest barrier to reconstructing these patterns likely lies in the small mammal record; taphonomic processes overwhelmingly bias against the preservation of small body sizes, which in North American mammals makes up the bulk of diversity. However, bird castings and other predatory castings offer an invaluable taphonomic window for preserving small mammal species, and one which is under-explored [87].

These results thus justify new avenues of research looking at the biogeographic response of biota to a variety of global change scenarios, including extinction events, in deep time. In particular, changes in beta diversity through time will help to determine the long-term processes that have sculpted present-day patterns in biogeography (especially in combination with palaeo-range reconstruction [8,11]), as well as help predict how we expect processes of community assembly and ecosystem function to change in response to ongoing anthropogenic disturbance (e.g. [88]). In concert, mapping the changing distribution of spatial richness

patterns potentially offers long-term data germane to ongoing efforts in conservation biology, for example, in designing the size and location of protected areas. At the broadest scale, this study joins the growing body of work illustrating that the fossil record represents a surprisingly faithful dataset recording the changing spatial distribution of biota over key intervals of Earth History [22–25].

Data accessibility. The datasets supporting this article have been uploaded as part of the electronic supplementary material. All R code and simulations results are available from the Dryad Digital Repository: https://doi.org/10.5061/dryad.wstqjq2jv [89].

Authors' contributions. S.A.F.D., D.F., and M.M.C. designed the research. S.A.F.D. and D.F. performed analyses. All authors contributed to writing the manuscript.

Competing interests. We declare we have no competing interests.

Funding. Funding was provided by an NSERC Discovery (grant no. RGPIN-2018-05305) and Canadian Museum of Nature Research Activity Grant to D.F. S.A.F.D. also acknowledges generous support from the Alexander von Humboldt Foundation, which is sponsored by the Federal Ministry for Education and Research in Germany.

Acknowledgements. S.A.F.D. would like to thank Dr Malu Jorge for helpful discussions. All authors would like to thank contributors to the Paleobiology Database, from which the fossil mammal localities used in this study were taken.

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
