## [Peer Review File · Proceedings of the Royal Society B: Biological Sciences]

Review History

RSPB-2020-0901.R0 (Original submission)

Review form: Reviewer 1

Recommendation

Accept with minor revision (please list in comments)

Scientific importance: Is the manuscript an original and important contribution to its field?

Good

General interest: Is the paper of sufficient general interest?

Good

Quality of the paper: Is the overall quality of the paper suitable?

Excellent

Is the length of the paper justified?

Yes

Should the paper be seen by a specialist statistical reviewer?

No

Do you have any concerns about statistical analyses in this paper? If so, please specify them explicitly in your report.

No

It is a condition of publication that authors make their supporting data, code and materials available - either as supplementary material or hosted in an external repository. Please rate, if applicable, the supporting data on the following criteria.

Is it accessible?

Yes

Is it clear?

Yes

Is it adequate?

Yes

Do you have any ethical concerns with this paper?

No

Comments to the Author

“The preservation potential of terrestrial biogeographic patterns” is an interesting paper that models the preservation potential of beta diversity patterns given a series of different taphonomic preservation scenarios and different numbers of sites. This paper formalizes some of the sensitivity tests that nearly every large-scale spatially explicit paleontological paper must perform before drawing conclusions, and it does so in an interesting and rigorous manner.

My main concern with this paper is the way in which it performs its beta diversity experiments. Understandably, when simulating extinctions, the authors kill off the x% of species with the smallest range sizes. This is, presumably, because it has been repeatedly demonstrated that species with small range sizes are most vulnerable to extinction. I think this is understandable, and I recognize the need to simplify a set of analyses in order to tell a concise and understandable story. However, there are a couple of issues with choosing to highlight this strategy with regard to these analyses:

- 1) As the authors note, the loss of small ranged species results in very minute changes in beta diversity, resulting in a very small effect size that the fossil reconstructions then have to try to capture. I am frankly amazed that they are able to detect this effect size at all using these subsampling techniques.
- 2) Given that the authors are considering the Cenozoic, there are many instances where the large-ranging species are the ones that go extinct, notably during the end-Pleistocene, but also at other times, as well.
- 3) Finally, the species, and therefore the biodiversity patterns, that are most detectable in the fossil record are rarely going to include these small-ranged species to begin with, particularly as we go back further in time. I recognize that the various size-biasing analyses were intended to demonstrate this. I'm just really wondering if the authors would be better served to highlight at least some extinction scenarios that result in a larger effect size in the change in biodiversity. Perhaps this is not possible, but I think it would improve the paper if it is.

I do appreciate that the authors performed a re-expansion analysis. I think it helps a bit with my above issues. However, it seems odd to have placed it entirely in the discussion section.

In general, I think this paper will be fairly well cited by spatial paleoecologists, and it may guide some of their sensitivity analyses. However, it won't preclude them having to do their own sensitivity analyses, as each situation is distinct.

I commend the authors on writing a very clear, easy to follow paper.

I have a few additional, minor comments below:

“pronounced biogeographic fabric” is very odd, metaphorical phrasing. I suggest the authors stick with “biogeographic patterns”.

The diversity re-mapping map in the conceptual figure makes it look like your method does a terrible job. Very nice conceptual figure, though. I’m a huge fan of every paper having a figure like this.

I found the figures a bit hard to follow without cross-referencing with the paper a fair bit. To me, this means that the authors need a bit more detail in the captions.

Figure 6 was particularly hard to interpret without referring to the text. The axes and labels are exactly the same and they are not distinguished in the caption even by letters.

Review form: Reviewer 2

Recommendation

Major revision is needed (please make suggestions in comments)

Scientific importance: Is the manuscript an original and important contribution to its field?

Good

General interest: Is the paper of sufficient general interest?

Acceptable

Quality of the paper: Is the overall quality of the paper suitable?

Good

Is the length of the paper justified?

Yes

Should the paper be seen by a specialist statistical reviewer?

No

Do you have any concerns about statistical analyses in this paper? If so, please specify them explicitly in your report.

No

It is a condition of publication that authors make their supporting data, code and materials available - either as supplementary material or hosted in an external repository. Please rate, if applicable, the supporting data on the following criteria.

Is it accessible?

N/A

Is it clear?

Yes

Is it adequate?

Yes

Do you have any ethical concerns with this paper?

No

Comments to the Author

This paper by Darroch et al. assesses how fossil sampling influences some biogeographic variables and whether incomplete preservation would allow the detection of the effects of an extinction event in these variables. I think this paper is very timely and I read it with great interest. Simulations are excellent research tools that allow us to see how various aspects of biogeography interact in controlled settings, without which the interpretation of observations remains dubious. I have a couple of issues with this paper, but with some modifications/changes it will be an useful contribution to Proceedings B.

(1) My biggest concern with the paper is its geographic scope. I believe that the IUCN range dataset is global, and so is the list of localities from the Pleistocene in the Paleobiology Database (although incomplete and the stratigraphic resolution varies). Global-scale, heterogeneous spatial sampling is probably our most important preservational problem when it comes to biogeographic-macroecological patterns. In my opinion, understanding how this global pattern of sampling influences the detectability of changes in alpha and beta diversity would greatly increase the relevance of the study.

I understand that although it would be preferable, going global would be a substantial change compared to the original analysis. If this is not feasible, can the authors perhaps extend the geographic scope to North America? Lots of readers are not from the US and when it comes to a globally relevant problem, such as extinction, natural boundaries should be preferred instead of administrative ones. In any case, limiting of the geographic scope requires justification.

(2) As the authors wrote in the discussion, there are reasons for removing species with narrower range sizes when simulating extinctions. However, as they also mention, there are also reasons to assume that some extinction triggers might select for traits that are unrelated to geographic range. I agree with the authors that future studies are necessary to assess the effects of traits on the extinction patterns. But given that random extinctions is the simplest scenario and it also represents another corner-case, I would argue that the paper would be greatly improved if range-independent extinctions could be added to the analyses in some form. In itself it is an interesting question to see whether the range size-bias in extinction probability has an effect on the beta diversity drop. Focusing more on the extinction process rather than the preservational processes would also make the paper more relevant to researchers that work with marine and/or invertebrate fossils.

(3) I had the impression that the implications of the results and their practical value for future research are not discussed in detail. For instance, how does our ability to detect the alpha/beta diversity aspect of a mass extinction influence the study of mass extinctions?

The choice of the taxonomic group and the spatial focus is justified with reference to the Pleistocene megafaunal extinction, yet there is barely any discussion on how these results actually help us better understand the biogeography of this event. Do the results allow us to guess about whether our observation under- or overestimate the actual biogeographic changes? The implications for present-day conservation is also mentioned in the introduction, but it is barely mentioned in the discussion.

I could not find any mention of a potential association between geographic range and body size. Given that body-size related taphonomic biases are suggested by the authors to be influencing the observed patterns of size-dependent extinction after fossilization, I think it would be interesting to assess whether an association between these two variables have relevance to the results.

The paper is altogether well-written. I only have some additional minor comments (line-by-line):

Title: To me the use of the adjective 'biogeographic' is too general (this applies to the whole manuscript), which makes the title somewhat misleading. There are many more biogeographic variables, that are not or barely mentioned in this paper, such as geodisparity and the partitioning to spatial units. Why not just say what the study actually tested: 'alpha and beta diversity patterns'?

Line 64: Some references to past attempts would be really useful.

Line 105: I would omit 'virtual' from this sentence. The species ranges are actual data, virtual species would imply that the ranges are procedurally generated.

Line 120: This sentence implies that pairwise metrics do not allow the partitioning to turnover and nestedness. Incorporating this aspect to the analyses would be interesting.

Line 127: 'overwhelmingly thought' is a bit strong. This implies universal prevalence for this pattern. Although extinction risk is indeed higher with narrower ranges, but not necessarily in every case.

Line 137: I suggest moving the reference to R to the end of the methods section.

Line 193: where does the 1% comes from?

Line 203: "adding a function" is really not necessary

Line 212: I would replace "Re-mapping" with "Reconstruction of", re-mapping is somewhat vague.

Line 216: The greek letter 'Tau' is consistently misspelled as 'Tao'

Line 219: Although the word 'Simmed' is perfectly alright in spoken language, I think 'Simulated' would be more appropriate in formal publications. This applies to Fig. 5 as well.

Line 287: Perhaps mention that disaster species can be newly originating ones.

Figure 1. Consider adding headers to the figure. The first column displays the extinctions scenarios, the second, the preservation scenarios, the last one is the result. Here you can also write reconstruction instead of re-mapping. Also consider using a different color-scheme (a blue-red, inverse heatmap, perhaps?) than the default in `raster::plot()` to make the plots more distinct.

Figure 2. Could you spell out the abbreviated names? Also consider changing the color of the no-extinction scenario as that serves as the baseline.

Decision letter (RSPB-2020-0901.R0)

08-Jun-2020

Dear Dr Darroch:

I am writing to inform you that your manuscript RSPB-2020-0901 entitled "The preservation potential of terrestrial biogeographic patterns" has, in its current form, been rejected for publication in Proceedings B.

This action has been taken on the advice of referees, who have recommended that substantial revisions are necessary. With this in mind we would be happy to consider a resubmission, provided the comments of the referees are fully addressed. However please note that this is not a provisional acceptance. Reviewers and the Associate Editor are supportive of the MS but require some substantial changes; not purely presentational.

Sincerely,
Dr John Hutchinson, Editor
mailto:proceedingsb@royalsociety.org

Associate Editor
Board Member: 1
Comments to Author:

Thank you for the opportunity to review this manuscript. Reviews from two referees have now been received. Both complimented the importance of the topic and appreciated the modeling approach. However, both also raised several issues with how many aspects of the simulations were conducted. Both Referees (especially Referee 1) expressed concerns about the way in which species range was used as a criterion for extinction in the models. In addition, Referee 2 questioned the chosen geographic scope of the analyses and recommended expanding them. Additional emphasis was also recommended to be placed on the implications of the study for the general study of mass extinctions as well as conservation in the present day. These topics would broaden the reach of the MS, but did not ultimately receive the attention that might have been expected from the Introduction to the paper.

Considering these issues, and the other specific points noted in the reviews, I cannot recommend this manuscript for publication in its current form. However, if you feel you can address the points noted in the referee reports, it may be possible to resubmit the manuscript for re-review. In a resubmission, please carefully consider the comments provided in the reviews, and explain how your revisions have addressed the concerns that were raised.

Thank you once again for your submission. I hope you find the referee comments to provide constructive guidance for revising your report on your study.

Reviewer(s)' Comments to Author:

Referee: 1

Comments to the Author(s)

"The preservation potential of terrestrial biogeographic patterns" is an interesting paper that models the preservation potential of beta diversity patterns given a series of different taphonomic preservation scenarios and different numbers of sites. This paper formalizes some of the sensitivity tests that nearly every large-scale spatially explicit paleontological paper must perform before drawing conclusions, and it does so in an interesting and rigorous manner.

My main concern with this paper is the way in which it performs its beta diversity experiments. Understandably, when simulating extinctions, the authors kill off the x% of species with the smallest range sizes. This is, presumably, because it has been repeatedly demonstrated that species with small range sizes are most vulnerable to extinction. I think this is understandable, and I recognize the need to simplify a set of analyses in order to tell a concise and understandable story. However, there are a couple of issues with choosing to highlight this strategy with regard to these analyses:

- 1) As the authors note, the loss of small ranged species results in very minute changes in beta diversity, resulting in a very small effect size that the fossil reconstructions then have to try to capture. I am frankly amazed that they are able to detect this effect size at all using these subsampling techniques.
- 2) Given that the authors are considering the Cenozoic, there are many instances where the large-ranging species are the ones that go extinct, notably during the end-Pleistocene, but also at other times, as well.
- 3) Finally, the species, and therefore the biodiversity patterns, that are most detectable in the fossil record are rarely going to include these small-ranged species to begin with, particularly as we go back further in time. I recognize that the various size-biasing analyses were intended to demonstrate this. I'm just really wondering if the authors would be better served to highlight at least some extinction scenarios that result in a larger effect size in the change in biodiversity. Perhaps this is not possible, but I think it would improve the paper if it is.

I do appreciate that the authors performed a re-expansion analysis. I think it helps a bit with my above issues. However, it seems odd to have placed it entirely in the discussion section.

In general, I think this paper will be fairly well cited by spatial paleoecologists, and it may guide some of their sensitivity analyses. However, it won't preclude them having to do their own sensitivity analyses, as each situation is distinct.

I commend the authors on writing a very clear, easy to follow paper.

I have a few additional, minor comments below:

"pronounced biogeographic fabric" is very odd, metaphorical phrasing. I suggest the authors stick with "biogeographic patterns".

The diversity re-mapping map in the conceptual figure makes it look like your method does a terrible job. Very nice conceptual figure, though. I'm a huge fan of every paper having a figure like this.

I found the figures a bit hard to follow without cross-referencing with the paper a fair bit. To me, this means that the authors need a bit more detail in the captions.

Figure 6 was particularly hard to interpret without referring to the text. The axes and labels are exactly the same and they are not distinguished in the caption even by letters.

Referee: 2

Comments to the Author(s)

This paper by Darroch et al. assesses how fossil sampling influences some biogeographic variables and whether incomplete preservation would allow the detection of the effects of an extinction event in these variables. I think this paper is very timely and I read it with great interest. Simulations are excellent research tools that allow us to see how various aspects of biogeography interact in controlled settings, without which the interpretation of observations remains dubious. I have a couple of issues with this paper, but with some modifications/changes it will be an useful contribution to Proceedings B.

(1) My biggest concern with the paper is its geographic scope. I believe that the IUCN range dataset is global, and so is the list of localities from the Pleistocene in the Paleobiology Database (although incomplete and the stratigraphic resolution varies). Global-scale, heterogeneous spatial sampling is probably our most important preservational problem when it comes to biogeographic-macroecological patterns. In my opinion, understanding how this global pattern of sampling influences the detectability of changes in alpha and beta diversity would greatly increase the relevance of the study.

I understand that although it would be preferable, going global would be a substantial change compared to the original analysis. If this is not feasible, can the authors perhaps extend the geographic scope to North America? Lots of readers are not from the US and when it comes to a globally relevant problem, such as extinction, natural boundaries should be preferred instead of administrative ones. In any case, limiting of the geographic scope requires justification.

(2) As the authors wrote in the discussion, there are reasons for removing species with narrower range sizes when simulating extinctions. However, as they also mention, there are also reasons to assume that some extinction triggers might select for traits that are unrelated to geographic range. I agree with the authors that future studies are necessary to assess the effects of traits on the extinction patterns. But given that random extinctions is the simplest scenario and it also represents another corner-case, I would argue that the paper would be greatly improved if range-independent extinctions could be added to the analyses in some form. In itself it is an interesting question to see whether the range size-bias in extinction probability has an effect on the beta diversity drop. Focusing more on the extinction process rather than the preservational processes would also make the paper more relevant to researchers that work with marine and/or invertebrate fossils.

(3) I had the impression that the implications of the results and their practical value for future research are not discussed in detail. For instance, how does our ability to detect the alpha/beta diversity aspect of a mass extinction influence the study of mass extinctions?

The choice of the taxonomic group and the spatial focus is justified with reference to the Pleistocene megafaunal extinction, yet there is barely any discussion on how these results actually help us better understand the biogeography of this event. Do the results allow us to guess about whether our observation under- or overestimate the actual biogeographic changes? The implications for present-day conservation is also mentioned in the introduction, but it is barely mentioned in the discussion.

I could not find any mention of a potential association between geographic range and body size. Given that body-size related taphonomic biases are suggested by the authors to be influencing the observed patterns of size-dependent extinction after fossilization, I think it would be interesting to assess whether an association between these two variables have relevance to the results.

The paper is altogether well-written. I only have some additional minor comments (line-by-line):

Title: To me the use of the adjective 'biogeographic' is too general (this applies to the whole manuscript), which makes the title somewhat misleading. There are many more biogeographic variables, that are not or barely mentioned in this paper, such as geodispersity and the partitioning to spatial units. Why not just say what the study actually tested: 'alpha and beta diversity patterns'?

Line 64: Some references to past attempts would be really useful.

Line 105: I would omit 'virtual' from this sentence. The species ranges are actual data, virtual species would imply that the ranges are procedurally generated.

Line 120: This sentence implies that pairwise metrics do not allow the partitioning to turnover and nestedness. Incorporating this aspect to the analyses would be interesting.

Line 127: 'overwhelmingly thought' is a bit strong. This implies universal prevalence for this pattern. Although extinction risk is indeed higher with narrower ranges, but not necessarily in every case.

Line 137: I suggest moving the reference to R to the end of the methods section.

Line 193: where does the 1% comes from?

Line 203: "adding a function" is really not necessary

Line 212: I would replace "Re-mapping" with "Reconstruction of", re-mapping is somewhat vague.

Line 216: The greek letter 'Tau' is consistently misspelled as 'Tao'

Line 219: Although the word 'Simmed' is perfectly alright in spoken language, I think 'Simulated' would be more appropriate in formal publications. This applies to Fig. 5 as well.

Line 287: Perhaps mention that disaster species can be newly originating ones.

Figure 1. Consider adding headers to the figure. The first column displays the extinctions scenarios, the second, the preservation scenarios, the last one is the result. Here you can also write reconstruction instead of re-mapping. Also consider using a different color-scheme (a blue-red, inverse heatmap, perhaps?) than the default in `raster::plot()` to make the plots more distinct.

Figure 2. Could you spell out the abbreviated names? Also consider changing the color of the no-extinction scenario as that serves as the baseline.

Author's Response to Decision Letter for (RSPB-2020-0901.R0)

See Appendix A.

RSPB-2020-2927.R0

Review form: Reviewer 1

Recommendation

Accept with minor revision (please list in comments)

Scientific importance: Is the manuscript an original and important contribution to its field?

Good

General interest: Is the paper of sufficient general interest?

Good

Quality of the paper: Is the overall quality of the paper suitable?

Good

Is the length of the paper justified?

Yes

Should the paper be seen by a specialist statistical reviewer?

No

Do you have any concerns about statistical analyses in this paper? If so, please specify them explicitly in your report.

No

It is a condition of publication that authors make their supporting data, code and materials available - either as supplementary material or hosted in an external repository. Please rate, if applicable, the supporting data on the following criteria.

Is it accessible?

Yes

Is it clear?

Yes

Is it adequate?

Yes

Do you have any ethical concerns with this paper?

No

Comments to the Author

This paper remains interesting and has been greatly improved due to its expanded analyses. The greatest weakness at this point are the many minor issues with the figures and captions.

The font in the figures is so small that I have to zoom in a lot to read it (especially on axes). There is plenty of room to expand font sizes throughout.

Fig. 1 caption refers to letters, but there are no letters in the figure.

Fig. 1: 'taphonomy', 'lagerstätten', and 'castings' are not mentioned in the caption, and it's unclear how these interact with the sampling schema from looking at the figure. 'Preservation scenarios' arrow flow is pretty unclear to me. 'random' does not flow into 'localities'. I think

either the logic of those arrows is not quite right or I don't understand the different types of arrows. The different types of arrows should be defined in the caption or in a key.

Fig. 2 dots on the "top row" are so small that I cannot tell a color difference.

Fig. 2 I like the parallel images from the conceptual figure being used here. The figure doesn't seem horizontally limited. Why not make them larger & place them to the right?

Fig. 2 The caption of this feels very disjointed and hard to read. It also seems odd to have a "top row" designation and then A-E. Why not just do A-F?

Fig. 2. I like the use of gray, blue, & orange. Are the boxplots showing quartiles & medians? Specify.

Fig. 2 Each figure should have a title that summarizes its purpose.

Fig. 3: "Kendall's Tau" was updated throughout the text but not in the figure.

Fig. 3: Why title the left column "simmed" in the figure & "simulated" in the caption? Use the more formal "simulated" throughout.

Fig. 3: refers to letters, but does not include them in the figure.

Fig. 4: write out "extinction". Make "random" "rand." if necessary

Review form: Reviewer 2

Recommendation

Accept with minor revision (please list in comments)

Scientific importance: Is the manuscript an original and important contribution to its field?

Good

General interest: Is the paper of sufficient general interest?

Good

Quality of the paper: Is the overall quality of the paper suitable?

Good

Is the length of the paper justified?

Yes

Should the paper be seen by a specialist statistical reviewer?

No

Do you have any concerns about statistical analyses in this paper? If so, please specify them explicitly in your report.

No

It is a condition of publication that authors make their supporting data, code and materials available - either as supplementary material or hosted in an external repository. Please rate, if applicable, the supporting data on the following criteria.

Is it accessible?

Yes

Is it clear?

Yes

Is it adequate?

Yes

Do you have any ethical concerns with this paper?

No

Comments to the Author

I reviewed this paper before, which I quite liked even without the modifications that the authors implemented since.

I found that the authors addressed my comments and questions adequately and that the text was appropriately amended.

I only have minor comments about the paper:

Line 81 - 'species richness' is redundant here. If species are homogeneously distributed, species richness will be as well.

Line 92 - "a simulations framework": simulation framework?

Line 109 - "the North America": I recommend dropping the article.

Line 112 - "Sorenson": are you sure that this is the right spelling? Sørensen or Sorensen is the usual way.

Line 115 - Since the partitioning to nestedness and evenness is not part of the study, I recommend dropping the reference to it. If deemed relevant, mention this in the discussion instead.

Line 131 - I like the way the numbers are associated with the experiment names. I recommend keeping this association consistent throughout the manuscript, i.e. always using the number and the name of the experiment together (for instance in the paragraph from line 249). In my opinion this would help the reader to better remember which description fits which name when she/he processes the results.

Line 143 - When referring to the Rancholabrean it would be helpful to give the approximate age in years. My guess is that many neontologists (or marine paleontologists) are not familiar with the nomenclature.

Line 153-157 - The equation on Line 157 comes from Western, 1975 too right? Should equations not be numbered? Please double check whether the equation referring matches the criteria of the journal. "Log" should not be capitalized, and please italicize mathematical variables throughout the text.

Line 167 - There is no function definition either here or above. Please refer to the equation throughout the paragraph or write a formal definition for the function, otherwise this whole section reads somewhat fuzzy.

Line 181 and 185 - instead of 'function' I recommend using 'preservation model' or 'process'.

Line 287-290 - This sentence is a bit difficult to understand, since it starts with gamma diversity and then end up with beta diversity. Could you rephrase this?

Line 318 - When talking about 'experiments' you do you mean 'preservation experiments'? The extinction modeling is already an experiment.

Line 324 - Please refer to a few of these studies here.

Line 325 - This is a very long sentence. I recommend inserting a full stop after 'body size'.

Line 430 - "Violins": Consider using "violin plots" instead.

Fig.1 - Consider dropping the boxes around the maps and drawing them around the portions representing preservation scenarios, instead - might result in a cleaner impression.

Fig. 3 - Labels on the figure mismatch the caption (I suppose this is an older version of the figure).

Fig. 4 - Note that the meaning of the word "baseline" is not the same in the caption and figure. I recommend renaming the column names from baseline and results to 'Complete preservation' and 'Experiment 5 - Casting'. Also please mention in the caption how many sites were used get the estimates.

Supplementary material: There is a larger chunk of text between ESM Fig. 3 and 4. Consider making a separate section from this material (e.g. supplementary text). Also, please edit the axis and panel labels for ESM 5.

Supplied code: I deeply value that the authors shared their final code. Unfortunately, I did not have the time to go through all of it and adjust the code so it would run on my computer.

The utility of this material could be enhanced by:

- structuring the files in relevant directories;
- increasing portability, i.e. decreasing the number of instances where code had to be manually edited to make it run;
- clearly indicating what the execution order of the code is to gain all results, and the purpose of individual scripts;
- and recording the versions of the used software packages and citing them as supplementary references.

Decision letter (RSPB-2020-2927.R0)

11-Jan-2021

Dear Dr Darroch:

Your manuscript has now been peer reviewed and the reviews have been assessed by an Associate Editor. The reviewers' comments (not including confidential comments to the Editor) and the comments from the Associate Editor are included at the end of this email for your reference. As you will see, the reviewers and the Editors have raised some concerns with your manuscript and we would like to invite you to revise your manuscript to address them.

Research ethics:

Use of animals and field studies:

It is a condition of publication that you make available the data and research materials supporting the results in the article (<https://royalsociety.org/journals/authors/author-guidelines/#data>). Datasets should be deposited in an appropriate publicly available repository and details of the associated accession number, link or DOI to the datasets must be included in the Data Accessibility section of the article (<https://royalsociety.org/journals/ethics-policies/data-sharing-mining/>). Reference(s) to datasets should also be included in the reference list of the article with DOIs (where available).

All supplementary materials accompanying an accepted article will be treated as in their final form. They will be published alongside the paper on the journal website and posted on the online

figshare repository. Files on figshare will be made available approximately one week before the accompanying article so that the supplementary material can be attributed a unique DOI. Please try to submit all supplementary material as a single file.

Please submit a copy of your revised paper within three weeks. If we do not hear from you within this time your manuscript will be rejected. If you are unable to meet this deadline please let us know as soon as possible, as we may be able to grant a short extension.

Best wishes,
Dr John Hutchinson, Editor
mailto:proceedingsb@royalsociety.org

Associate Editor Board Member

Comments to Author:

Thank you for submitting your revised manuscript. Both original referees complemented the work that was performed to address the recommendations that were made, including the new and revised analyses that were conducted. Both referees also identified some remaining points that would benefit from clarification or adjustments. Referee 1 detailed several items in the figures and their captions that could be clarified or corrected. In addition, Referee 2 detailed several specific points in the text, figures, supplementary material, as well as suggestions to improve the supplied code. In addition to these points, one further correction I would recommend is for the sentence in L341, where it would help to use a semicolon in place of the comma before “however”, or divide this sentence into two sentences for clarity.

With these recommendations, I encourage you to submit a revised version of the manuscript that addresses the comments noted above. Thank you once again for your submission.

Reviewer(s)' Comments to Author:

Referee: 1

Comments to the Author(s).

This paper remains interesting and has been greatly improved due to its expanded analyses. The greatest weakness at this point are the many minor issues with the figures and captions.

The font in the figures is so small that I have to zoom in a lot to read it (especially on axes). There is plenty of room to expand font sizes throughout.

Fig. 1 caption refers to letters, but there are no letters in the figure.

Fig. 1: ‘taphonomy’, ‘lagerstätten’, and ‘castings’ are not mentioned in the caption, and it’s unclear how these interact with the sampling schema from looking at the figure. ‘Preservation scenarios’ arrow flow is pretty unclear to me. ‘random’ does not flow into ‘localities’. I think either the logic of those arrows is not quite right or I don’t understand the different types of arrows. The different types of arrows should be defined in the caption or in a key.

Fig. 2 dots on the “top row” are so small that I cannot tell a color difference.

Fig. 2 I like the parallel images from the conceptual figure being used here. The figure doesn't seem horizontally limited. Why not make them larger & place them to the right?

Fig. 2 The caption of this feels very disjointed and hard to read. It also seems odd to have a "top row" designation and then A-E. Why not just do A-F?

Fig. 2. I like the use of gray, blue, & orange. Are the boxplots showing quartiles & medians? Specify.

Fig. 2 Each figure should have a title that summarizes its purpose.

Fig. 3: "Kendall's Tau" was updated throughout the text but not in the figure.

Fig. 3: Why title the left column "simmed" in the figure & "simulated" in the caption? Use the more formal "simulated" throughout.

Fig. 3: refers to letters, but does not include them in the figure.

Fig. 4: write out "extinction". Make "random" "rand." if necessary

Referee: 2

Comments to the Author(s).

I reviewed this paper before, which I quite liked even without the modifications that the authors implemented since.

I found that the authors addressed my comments and questions adequately and that the text was appropriately amended.

I only have minor comments about the paper:

Line 81 - 'species richness' is redundant here. If species are homogeneously distributed, species richness will be as well.

Line 92 - "a simulations framework": simulation framework?

Line 109 - "the North America": I recommend dropping the article.

Line 112 - "Sorenson": are you sure that this is the right spelling? Sørensen or Sorensen is the usual way.

Line 115 - Since the partitioning to nestedness and evenness is not part of the study, I recommend dropping the reference to it. If deemed relevant, mention this in the discussion instead.

Line 131 - I like the way the numbers are associated with the experiment names. I recommend keeping this association consistent throughout the manuscript, i.e. always using the number and the name of the experiment together (for instance in the paragraph from line 249). In my opinion this would help the reader to better remember which description fits which name when she/he processes the results.

Line 143 - When referring to the Rancholabrean it would be helpful to give the approximate age in years. My guess is that many neontologists (or marine paleontologists) are not familiar with the nomenclature.

Line 153-157 - The equation on Line 157 comes from Western, 1975 too right? Should equations not be numbered? Please double check whether the equation referring matches the criteria of the

journal. "Log" should not be capitalized, and please italicize mathematical variables throughout the text.

Line 167 - There is no function definition either here or above. Please refer to the equation throughout the paragraph or write a formal definition for the function, otherwise this whole section reads somewhat fuzzy.

Line 181 and 185 - instead of 'function' I recommend using 'preservation model' or 'process'.

Line 287-290 - This sentence is a bit difficult to understand, since it starts with gamma diversity and then end up with beta diversity. Could you rephrase this?

Line 318 - When talking about 'experiments' you do you mean 'preservation experiments'? The extinction modeling is already an experiment.

Line 324 - Please refer to a few of these studies here.

Line 325 - This is a very long sentence. I recommend inserting a full stop after 'body size'.

Line 430 - "Violins": Consider using "violin plots" instead.

Fig.1 - Consider dropping the boxes around the maps and drawing them around the portions representing preservation scenarios, instead - might result in a cleaner impression.

Fig. 3 - Labels on the figure mismatch the caption (I suppose this is an older version of the figure).

Fig. 4 - Note that the meaning of the word "baseline" is not the same in the caption and figure. I recommend renaming the column names from baseline and results to 'Complete preservation' and 'Experiment 5 - Casting'. Also please mention in the caption how many sites were used get the estimates.

Supplementary material: There is a larger chunk of text between ESM Fig. 3 and 4. Consider making a separate section from this material (e.g. supplementary text). Also, please edit the axis and panel labels for ESM 5.

Supplied code: I deeply value that the authors shared their final code. Unfortunately, I did not have the time to go through all of it and adjust the code so it would run on my computer.

The utility of this material could be enhanced by:

- structuring the files in relevant directories;
- increasing portability, i.e. decreasing the number of instances where code had to be manually edited to make it run;
- clearly indicating what the execution order of the code is to gain all results, and the purpose of individual scripts;
- and recording the versions of the used software packages and citing them as supplementary references.

Author's Response to Decision Letter for (RSPB-2020-2927.R0)

See Appendix B.

Decision letter (RSPB-2020-2927.R1)

31-Jan-2021

Dear Dr Darroch

I am pleased to inform you that your manuscript entitled "The preservation potential of terrestrial biogeographic patterns" has been accepted for publication in Proceedings B. Congratulations!!

Open Access

Paper charges

Sincerely,

Dr John Hutchinson

Associate Editor:

Board Member

Comments to Author:

Thank you for submitting the revised version of your manuscript, which has thoroughly addressed the comments raised by the referees. Your contribution to Proceedings B is appreciated.

Appendix A

Response to reviewers document

Reviewer #1

1) My main concern with this paper is the way in which it performs its beta diversity experiments. Understandably, when simulating extinctions, the authors kill off the x% of species with the smallest range sizes. This is, presumably, because it has been repeatedly demonstrated that species with small range sizes are most vulnerable to extinction. I think this is understandable, and I recognize the need to simplify a set of analyses in order to tell a concise and understandable story. However, there are a couple of issues with choosing to highlight this strategy with regard to these analyses. As the authors note, the loss of small ranged species results in very minute changes in beta diversity, resulting in a very small effect size that the fossil reconstructions then have to try to capture. I am frankly amazed that they are able to detect this effect size at all using these subsampling techniques.

Given that the authors are considering the Cenozoic, there are many instances where the large-ranging species are the ones that go extinct, notably during the end-Pleistocene, but also at other times, as well.

CHANGES MADE: Although we think that this scenario is ultimately quite unlikely – we argue that end-Pleistocene extinction selected against taxa with other life history traits, some of which are only loosely correlated with body size and range size – we agree that this is a useful exercise, and have repeated our ‘castings’ experiment (now using mammal distributions across all of North America; see our response to comment #7), only this time removing 25%, 50% and 75% of the largest-ranged species. The results are interesting; apart from producing a very different pattern of ‘baseline’ results, calculating beta diversity after applying our stepped series of locality- and taphonomy-based filters reveals that these hypothetical extinction thresholds will likely be recorded as an increase, rather than decrease in beta diversity in the fossil data.

ESM 4 – Simulated beta diversity for an additional ‘castings’ experiment that simulates the extinction of large-, rather than small-ranged species. Baseline (i.e., 0% extinction) scenario illustrated in grey. Boxplots colored blue indicate those that overlap with the baseline and thus are unlikely to be detectable in the fossil record; those colored orange possess no overlap with the baseline, and thus indicate scenarios where change in beta diversity is most likely detectable. These simulations suggest that removing large-ranged species may be recorded as significant increase in beta diversity, when the actual (i.e., baseline) scenario instead records decrease.

Although this result is surprising, this fundamental difference in observed vs. actual changes in beta diversity are best-explained in terms of both the distribution of sampling sites in baseline vs. experimental scenarios, and the sizes of ranges being removed within increasing extinction intensity. However, both because this mechanism requires a lengthy explanation, and because we consider extinction events selecting against large-ranged taxa unlikely (see explanation in revised ESM document), we have put Figures and text describing this series of simulations experiments in the Supplementary information for now. We do, however, reference these experiments in the main document, and make the point that a future suite of experiments should investigate the biogeographic signal of extinction events that select against other, non-range-based life history traits.

“Lastly, although the vast majority of studies have identified small range size as a predictor of extinction risk, some studies have suggested stronger correlations with other macroecological attributes, such as body size; future studies should therefore further investigate extinction risk factors other than geographic range, including life history traits [74]. In addition, a preliminary investigation into the preservation potential of beta diversity patterns when large-ranged species are driven extinct reveals some interesting contradictions, although we consider this scenario unlikely in the real world (ESM 4)” (line 322).

2) Finally, the species, and therefore the biodiversity patterns, that are most detectable in the fossil record are rarely going to include these small-ranged species to begin with, particularly as we go back further in time. I recognize that the various size-biasing analyses were intended to demonstrate this. I’m just really wondering if the authors would be better served to highlight at least some extinction scenarios that result in a larger effect size in the change in biodiversity. Perhaps this is not possible, but I think it would improve the paper if it is.

I do appreciate that the authors performed a re-expansion analysis. I think it helps a bit with my above issues. However, it seems odd to have placed it entirely in the discussion section.

CHANGES MADE: We agree; we have expanded this analysis and moved it to a more traditional position within the manuscript (i.e., introduced and justified in the methods). We now include both the re-expansion analysis and a new ‘null’ analysis (whereby extinction selectivity is random with respect to range size; see comment #9) as new additional experiments performed using our ‘castings’ scenario, as a means for testing the sensitivities of detection thresholds in different extinction selectivities.

3) “pronounced biogeographic fabric” is very odd, metaphorical phrasing. I suggest the authors stick with “biogeographic patterns”.

CHANGES MADE: We agree; ‘biogeographic fabric’ has been changed to ‘biogeography’ throughout.

4) The diversity re-mapping map in the conceptual figure makes it look like your method does a terrible job. Very nice conceptual figure, though.

CHANGES MADE: We thank the reviewer for the comment! Having re-performed analyses at a broader geographic scale (i.e., North America as opposed to the contiguous United States), we believe the diversity reconstruction given in our revised Figure 1 now constitutes a better-looking example.

6) I found the figures a bit hard to follow without cross-referencing with the paper a fair bit. To me, this means that the authors need a bit more detail in the captions.

CHANGES MADE: We apologize; we have reworded the captions, wherever possible specifically highlighting what each figure is supposed to demonstrate. We would be happy to modify these further if the reviewer thinks it necessary.

7) Figure 6 was particularly hard to interpret without referring to the text. The axes and labels are exactly the same and they are not distinguished in the caption even by letters.

CHANGES MADE: We agree; we have tried to make this figure clearer. We have added lettering to the figure, and reworded the caption as follows:

“Spatial correlations (as Kendall’s Tau) between actual and simulated richness maps for our 5 experiments illustrated as violin plots, both grid-wise (‘Simulated’ - left), and (right) ‘Extrapolated’ where a 3x3 grid cell moving window approach has been employed. Higher Tau values indicate better agreement between actual and simulated richness maps. Results of experiments ‘random’, ‘real localities’, ‘taphonomy’, ‘lagerstätten’ and ‘castings’ given in rows A-E respectively. Violins illustrate the distribution of correlations after 100 iterations in each experiment”

Reviewer #2:

8) My biggest concern with the paper is its geographic scope. I believe that the IUCN range dataset is global, and so is the list of localities from the Pleistocene in the Paleobiology Database (although incomplete and the stratigraphic resolution varies). Global-scale, heterogeneous spatial sampling is probably our most important preservational problem when it comes to biogeographic-macroecological patterns. In my opinion, understanding how this global pattern of sampling influences the detectability of changes in alpha and beta diversity would greatly increase the relevance of the study. I understand that although it would be preferable, going global would be a substantial change compared to the original analysis. If this is not feasible, can the authors perhaps extend the geographic scope to North America? Lots of readers are not from the US and when it comes to a globally relevant problem, such as extinction, natural boundaries should be preferred instead of administrative ones. In any case, limiting of the geographic scope requires justification.

CHANGES MADE: We agree, and thank the Reviewer for the suggestion. We have expanded the geographic scope of our analyses to North America (i.e., the United States, Canada, and Mexico); this also increases our species pool from 341 to 374 species. The results (see revised Figures 1-3) illustrate similar, but stronger, patterns to the ones described in the original manuscript.

9) As the authors wrote in the discussion, there are reasons for removing species with narrower range sizes when simulating extinctions. However, as they also mention, there are also reasons to assume that some extinction triggers might select for traits that are unrelated to geographic range. I agree with the authors that future studies are necessary to assess the effects of traits on the extinction patterns. But given that random extinctions is the simplest scenario and it also

represents another corner-case, I would argue that the paper would be greatly improved if range-independent extinctions could be added to the analyses in some form. In itself it is an interesting question to see whether the range size-bias in extinction probability has an effect on the beta diversity drop. Focusing more on the extinction process rather than the preservational processes would also make the paper more relevant to researchers that work with marine and/or invertebrate fossils.

CHANGES MADE: We agree; we now include a ‘null’ experiment, whereby extinction selectivity is random with respect to range size, and introduce this in the main manuscript alongside another additional experiment that explores the effect of range expansion among surviving taxa. The results are both interesting and re-assuring (see new Figure 5 reproduced below); they illustrate that random extinction does not, overall, produce consistent increases or decreases in beta diversity – a signal which is faithfully replicated in experiments. The large variance in simulated beta at the 75% extinction threshold does suggest that there may be a slight risk of false positive beta diversity signals at high extinction intensities (and when selectivity is random with respect to range size), but also that this is unlikely to be a pervasive or secular issue through longer swathes of geological time.

Modified from our new Figure 4: Simulated beta diversity for an additional experiment performed on null scenario, where extinction is random with respect to range size. Baseline results in left panel, with the results of our ‘castings’ experiment run using this scenario in right panel. The results illustrate that random extinction does not, overall, produce consistent increases or decreases in beta diversity – a signal which is faithfully replicated in experiments.

10) I had the impression that the implications of the results and their practical value for future research are not discussed in detail. For instance, how does our ability to detect the alpha/beta diversity aspect of a mass extinction influence the study of mass extinctions? The choice of the taxonomic group and the spatial focus is justified with reference to the Pleistocene megafaunal extinction, yet there is barely any discussion on how these results actually help us better understand the biogeography of this event. Do the results allow us to guess about whether our observation under- or overestimate the actual biogeographic changes? The implications for present-day conservation is also mentioned in the introduction, but it is barely mentioned in the discussion.

CHANGES MADE: We agree; although we emphasize that the main thrust of this study is to establish whether reconstructing biogeographic patterns from the terrestrial fossil record is a

valid approach (rather than reconstructing or re-interpreting said patterns from specific intervals in Earth history), we agree that some extra text is needed in the Discussion section that emphasizes the potential utility of these data. We have therefore added a short paragraph to the end of the Discussion section that highlights these points:

“These results thus justify new avenues of research looking at the biogeographic response of biota to a variety of global change scenarios, including extinction events, in deep time. In particular, changes in beta diversity through time will help to determine the long-term processes that have sculpted present-day patterns in biogeography (especially in combination with paleo-range reconstruction [8,11]), as well as help predict how we expect processes of community assembly and ecosystem function to change in response to ongoing anthropogenic disturbance (e.g., [86]). In concert, mapping the changing distribution of spatial richness patterns potentially offers long-term data germane to ongoing efforts in conservation biology, for example, in designing the size and location of protected areas. At the broadest scale, this study joins the growing body of work illustrating that the fossil record represents a surprisingly faithful dataset recording the changing spatial distribution of biota over key intervals of Earth History [22-25]” (line 369).

In terms of specific lessons for understanding the biogeography of late Pleistocene extinction (including beta diversity), to the best of our knowledge, no focused study dealing with beta diversity and/or spatial richness patterns has yet been published on this interval, and so we are reluctant to speculate. However, one of the authors listed here has recently performed an analysis along these lines (Fraser et al., in review); if these data are published in short order, then we would be happy to extend our discussion to include an interpretation of these results in light of our simulations.

11) I could not find any mention of a potential association between geographic range and body size. Given that body-size related taphonomic biases are suggested by the authors to be influencing the observed patterns of size-dependent extinction after fossilization, I think it would be interesting to assess whether an association between these two variables has relevance to the results.

CHANGES MADE: We agree; although the relationship between body size and range size is only weakly positive in our data (see figure below), and thus we would expect the results to broadly resemble those of the null experiments described under comment #9, body size is a life-history attribute that has been associated with extinction risk in the relatively recent geological past (and one that possesses a loose correlation with range size). To address this, we have performed an additional experiment that preferentially removes the largest-ranged (rather than smallest-ranged) species. Although the results of these experiments are described and discussed in the supplementary material, we refer to them in the main manuscript, and also make the point that a future suite of experiments should investigate the biogeographic signal of extinction events that select against other, non-range-based life history traits – see our response to comment #1.

Plot illustrating the weak positive relationship between log range size and log body size for the 374-mammal species used in our experiments; linear regression through these data given in blue ($R^2 = 0.016$; $p = 0.009$).

12) Title: To me the use of the adjective ‘biogeographic’ is too general (this applies to the whole manuscript), which makes the title somewhat misleading. There are many more biogeographic variables, that are not or barely mentioned in this paper, such as geodisparity and the partitioning to spatial units. Why not just say what the study actually tested: ‘alpha and beta diversity patterns’?

CHANGES NOT MADE: We respectfully disagree with the reviewer here; although it is doubtless true that there are innumerable facets of biogeography that are not tested in our experiments, from a paleontological perspective beta diversity and spatial patterns of richness (i.e., biodiversity hotspots) are among the most fundamental, and arguably at the limit of what workers would realistically attempt to reconstruct from fossil data. Consequently, we feel as though our title will resonate with paleontologists and ecologists alike, and moreover will help interested readers to find the paper. Follow-up work targeting the preservation potential of more specific facets of biogeography will accordingly be assigned more specific titles.

13) Line 64: Some references to past attempts would be really useful.

CHANGES MADE: We agree; references have been added to this sentence.

14) Line 105: I would omit ‘virtual’ from this sentence. The species ranges are actual data, virtual species would imply that the ranges are procedurally generated.

CHANGES MADE: This sentence has been reworded as follows:

“In order to preserve realistic range geometries and size distributions in simulations, we use the polygon distributional data for 374 extant terrestrial mammal species (taken from the IUCN Redlist: www.iucnredlist.org/) whose ranges extend into North America (i.e., the United States, Canada, and Mexico); these species describe an approximately log-normal distribution of range

sizes (ESM 1), which is typical for a majority of terrestrial taxonomic groups [44-47]” (line 105).

15) Line 120: This sentence implies that pairwise metrics do not allow the partitioning to turnover and nestedness. Incorporating this aspect to the analyses would be interesting.

CHANGES MADE: In order to create space for additional analyses (see our responses to comments #1, 2, 9 and 11), we have removed all text pertaining to pairwise beta metrics.

16) Line 127: ‘overwhelmingly thought’ is a bit strong. This implies universal prevalence for this pattern. Although extinction risk is indeed higher with narrower ranges, but not necessarily in every case.

CHANGES MADE: We agree; this sentence now reads:

“First, extinction events are typically thought to select against small-ranged taxa (leaving more species common to all local species pools and thus reducing beta diversity)” (line 216).

17) Line 137: I suggest moving the reference to R to the end of the methods section.

CHANGES MADE: We agree; the reference has been moved.

18) Line 193: where does the 1% comes from?

CHANGES MADE: We have used this estimate based on the relative rarity of fossil lagerstätten in the known fossil record. In any given time period, the number of lagerstätten varies from zero to ~100 (Muscente et al. 2017). Fossil lagerstätten are rare because the incredible preservation with which they are typically associated requires very specific preservational and depositional processes. Globally, for the Cenozoic, Muscente et al. (2017) report between ~5 and ~60 fossil lagerstätten. Of the total number of fossil localities reported in the Paleobiology Database for the Cenozoic (52,967), these lagerstätten therefore represent between 0.001% and 0.1% of Cenozoic sites. However, most of these lagerstätten appear concentrated in the Neogene, which encompasses the Pleistocene. The total number of localities reported in the PBDB for the Neogene is 19,171. Therefore, ~60 fossil lagerstätten would represent 0.3% of sites. Our 1% estimate is therefore high. But we consider caves to be lagerstätten and caves are comparatively common during the Pleistocene. The Neotoma database, which is North American Centric, reports 4,529 sites of which middens comprise 40 (~1%) and sites containing the name “cave” comprise 455 (~10%). The latter number is, however, likely to be an overestimate, given that Neotoma does not provide a measure of the degree of preservation at different cave sites and so many are likely not lagerstätten. Some brief text to support this point has been added to the manuscript:

“To do this, we include another function which introduces a 1% probability that any simulated fossil locality is characterized by exceptional preservation, and thus preserves every species that occurs there. This 1% figure reflects a conservative estimate for the rarity of fossil lagerstätten [59]” (line 181)

19) Line 203: “adding a function” is really not necessary

CHANGES MADE: The text now reads:

“In our fifth experiment (5 – ‘castings’), therefore, we simulate the collection of bird castings from all localities by recording all species between 5-800g body mass (which represents a typical range of prey size for medium-sized owls [66]) as ‘found’” (line 197).

20) Line 212: I would replace “Re-mapping” with “Reconstruction of”, re-mapping is somewhat vague.

CHANGES MADE: We agree; we have changed ‘re-mapping’ to ‘reconstructing’ throughout (see also our response to point 26).

21) Line 216: The greek letter ‘Tau’ is consistently misspelled as ‘Tao’

CHANGES MADE: We apologize – this was probably an autocorrect issue. ‘Tao’ has been changed to ‘Tau’ throughout.

22) Line 219: Although the word ‘Simmed’ is perfectly alright in spoken language, I think ‘Simulated’ would be more appropriate in formal publications. This applies to Fig. 5 as well.

CHANGES MADE: We agree; ‘simmed’ has been changed to ‘simulated’ throughout.

23) Line 287: Perhaps mention that disaster species can be newly originating ones.

CHANGES MADE: We agree; this section has been moved to the methods section, and now reads:

“However, surviving species, or new species evolving in the aftermath, can experience an increase in range size as they proliferate and disperse in response to the availability of free ecospace [67-68]. Range expansion among surviving taxa would add more species common to all local assemblages, and thus exacerbate reductions in beta diversity” (line 217).

24) Figure 1. Consider adding headers to the figure. The first column displays the extinctions scenarios, the second, the preservation scenarios, the last one is the result. Here you can also write reconstruction instead of re-mapping. Also consider using a different color-scheme (a blue-red, inverse heatmap, perhaps?) than the default in `raster::plot()` to make the plots more distinct.

CHANGES MADE: We thank the reviewer for the suggestions; we have modified the figure as requested. We have added headers to the three ‘columns’ in the figure, and changed ‘re-mapping’ to ‘reconstruction’ (here and elsewhere in the manuscript). We have not, however, changed the color scheme; rather than being the default for `raster::plot()`, this is a custom white-forest green-yellow-orange color palette that we settled on after a lengthy period of experimentation. We are open to changing the color scheme, but for the moment feel that this scheme best illustrates the richness patterns.

25) Figure 2. Could you spell out the abbreviated names? Also consider changing the color of the no-extinction scenario as that serves as the baseline.

CHANGES MADE: We again thank the reviewer for the suggestion; in the need to save space, we have combined this figure with the following figure (results). In this revised figure we now spell out the abbreviated name in the y-axis labels, and have changed the colors of the 0% extinction scenarios to light grey.

Appendix B

Associate Editor

1.) Both original referees complemented the work that was performed to address the recommendations that were made, including the new and revised analyses that were conducted. Both referees also identified some remaining points that would benefit from clarification or adjustments. Referee 1 detailed several items in the figures and their captions that could be clarified or corrected. In addition, Referee 2 detailed several specific points in the text, figures, supplementary material, as well as suggestions to improve the supplied code.

CHANGES MADE: Thank-you – we have made all the changes suggested, while also trying to keep the manuscript to an appropriate length.

2.) In addition to these points, one further correction I would recommend is for the sentence in L341, where it would help to use a semicolon in place of the comma before “however”, or divide this sentence into two sentences for clarity.

CHANGES MADE: A semicolon has been added as the editor suggests.

Referee 1

3.) This paper remains interesting and has been greatly improved due to its expanded analyses. The greatest weakness at this point are the many minor issues with the figures and captions.

CHANGES MADE: We thank the reviewer – we have made all the suggested modifications, and hope the figures and captions are now fixed.

4.) The font in the figures is so small that I have to zoom in a lot to read it (especially on axes). There is plenty of room to expand font sizes throughout.

CHANGES MADE: Apologies; we have substantially increased the font size in all figures, both in the main text, and in the electronic supplementary material.

5.) Fig. 1 caption refers to letters, but there are no letters in the figure.

CHANGES MADE: Apologies; we have re-worded the caption for Figure 1.

6.) Fig. 1: ‘taphonomy’, ‘lagerstätten’, and ‘castings’ are not mentioned in the caption, and it’s unclear how these interact with the sampling schema from looking at the figure. ‘Preservation scenarios’ arrow flow is pretty unclear to me. ‘random’ does not flow into ‘localities’. I think either the logic of those arrows is not quite right or I don’t understand the different types of arrows. The different types of arrows should be defined in the caption or in a key.

CHANGES MADE: We agree; we have tried to simplify this figure by restricting ourselves to one type of arrow, and re-arranging the existing arrows in a way that hopefully makes the meaning more intuitive.

7.) Fig. 2 dots on the “top row” are so small that I cannot tell a color difference.

CHANGES MADE: The size of these points has been doubled.

8.) Fig. 2 I like the parallel images from the conceptual figure being used here. The figure doesn’t seem horizontally limited. Why not make them larger & place them to the right?

CHANGES MADE: We agree; we have expanded and moved these images; we hope this gives the figure a better balance.

9.) Fig. 2 The caption of this feels very disjointed and hard to read. It also seems odd to have a “top row” designation and then A-E. Why not just do A-F?

CHANGES MADE: We agree; we have re-labelled the panel rows and adjusted the text of the caption.

10.) Fig. 2. I like the use of gray, blue, & orange. Are the boxplots showing quartiles & medians? Specify.

CHANGES MADE: That is correct – we now specify this information in the caption.

11.) Fig. 2 Each figure should have a title that summarizes its purpose.

CHANGES MADE: We agree; we have added a brief sentence to the beginning of each figure caption in line with what the reviewer suggests.

12.) Fig. 3: “Kendall’s Tau” was updated throughout the text but not in the figure.

CHANGES MADE: Apologies – we have fixed this.

13.) Fig. 3: Why title the left column “simmed” in the figure & “simulated” in the caption? Use the more formal “simulated” throughout.

CHANGES MADE: Apologies – we have fixed this.

14.) Fig. 3: refers to letters, but does not include them in the figure.

CHANGES MADE: We apologize – we think that author SAFD may have inadvertently uploaded an older version of this figure with the revised manuscript. Either way, panel rows in this figure now have letters attached.

15.) Fig. 4: write out “extinction”. Make “random” “rand.” if necessary

CHANGES MADE: We agree; we have modified this label to read, ‘random extinction’.

Referee 2

16.) Line 81 - 'species richness' is redundant here. If species are homogeneously distributed, species richness will be as well.

CHANGES MADE: We agree; we have reworded this sentence and thank the reviewer for the suggestion:

“The fact that species are not distributed homogeneously in space is a fundamental observation in ecology, with huge efforts dedicated to determining the biotic, abiotic, and historical controls on richness patterns at a broad range of spatial scales [35]” (line 81).

17.) Line 92 - "a simulations framework": simulation framework?

CHANGES MADE: We agree; we have changed ‘simulations’ in this sentence to ‘simulation-based’.

18.) Line 109 - "the North America": I recommend dropping the article.

CHANGES MADE: Agreed! 'The' has been removed.

19.) Line 112 - "Sorenson": are you sure that this is the right spelling? Sørensen or Sorensen is the usual way.

CHANGES MADE: The reviewer is correct – we apologize. 'Sorenson' has been changed to 'Sørensen' throughout the manuscript and figures.

20.) Line 115 - Since the partitioning to nestedness and evenness is not part of the study, I recommend dropping the reference to it. If deemed relevant, mention this in the discussion instead.

CHANGES MADE: We agree; this sentence now reads:

"Multisite metrics (such as Sørensen's dissimilarity) are similar to Whittaker's [27] original formulation, and account for compositional heterogeneity for assemblages of more than two sites [48-49]" (line 114).

21.) Line 131 - I like the way the numbers are associated with the experiment names. I recommend keeping this association consistent throughout the manuscript, i.e. always using the number and the name of the experiment together (for instance in the paragraph from line 249). In my opinion this would help the reader to better remember which description fits which name when she/he processes the results.

CHANGES MADE: We agree – we have added numbers to descriptions of experiments wherever appropriate (with the exception of the section on 'Additional tests', where we fear it would slow down the narrative and become repetitive).

22.) Line 143 - When referring to the Rancholabrean it would be helpful to give the approximate age in years. My guess is that many neontologists (or marine paleontologists) are not familiar with the nomenclature.

CHANGES MADE: We agree; this sentence now reads:

"...fossil localities from the Rancholabrean North American Land Mammal Age (late Pleistocene – 240-11 kya)" (line 149).

23.) Line 153-157 - The equation on Line 157 comes from Western, 1975 too right? Should equations not be numbered? Please double check whether the equation referring matches the criteria of the journal. "Log" should not be capitalized, and please italicize mathematical variables throughout the text.

CHANGES MADE: We agree; we have removed the capital letters from 'log', and numbered equations in the text (although we were unable to find specific formatting instructions for these on the journal website). Mathematical variables (including 'Tau') have been italicized in the text. The equation the reviewer refers to is actually derived from Behrensmeyer et al. (1979) - although we derived population size using the Damuth (1981) equation, which was not done in the Behrensmeyer et al. study.

24.) Line 167 - There is no function definition either here or above. Please refer to the equation throughout the paragraph or write a formal definition for the function, otherwise this whole section reads somewhat fuzzy.

CHANGES MADE: For clarity we have replaced the term 'function' with "series of equations".

25.) Line 181 and 185 - instead of 'function' I recommend using 'preservation model' or 'process'.

CHANGES MADE: We agree; as above, we have replaced function with "series of equations".

26.) Line 287-290 - This sentence is a bit difficult to understand, since it starts with gamma diversity and then end up with beta diversity. Could you rephrase this?

CHANGES MADE: We agree; we suggest a simple fix could be to refer to gamma diversity later on in the sentence, and place some of the extra text in parentheses. The new wording is copied below, but we are happy to re-phrase again if the reviewer thinks this is unclear.

"Plotting estimates of gamma diversity (i.e. total North American diversity) over the same set of simulations (ESM 3) shows that, even when an extinction signal is captured in gamma (i.e., the overall number of recovered species), a decrease in beta diversity is not always detected" (line 319)

27.) Line 318 - When talking about 'experiments' you do you mean 'preservation experiments'? The extinction modeling is already an experiment.

CHANGES MADE: We have changed 'experiments' in this context to 'simulation experiments', and have reworded the beginning of this section in a way which hopefully makes it clearer:

"We acknowledge, however, that several aspects of our study are idealized. For example, in simulating extinction events of varying intensities, we remove the 25%, 50% and 75% smallest ranged taxa. Although past biotic crises have shown a general tendency to preferentially select against small-ranged species (e.g., [54]), this tendency is rarely perfect, and over some extinction events selectivity was more or less random with respect to geographic range size [53,72-73]. Our second additional test provides some insight into the potential impact of this different selectivity scenario. Baseline results for this test illustrate that random extinction does not produce consistent increases or decreases in beta diversity (Figure 4b). This is faithfully replicated in our simulation experiments, which likewise illustrate no consistent tendency for beta diversity to change with increasing extinction intensity" (line 342).

28.) Line 324 - Please refer to a few of these studies here.

CHANGES MADE: We have added citations to two key papers that deal with body mass as a correlate for extinction risk; these are given below:

[74] Smith FA, Smith REE, Lyons SK and Payne JL. 2018. Body size downgrading of mammals over the late Quaternary. *Science* 360: 310-313.

[75] Lyons SK, Smith FA and Brown JH. 2004. Of mice, mastodons and men: human-mediated extinctions on four continents. *Evolutionary Ecology Research* 6: 339-358.

29.) Line 325 - This is a very long sentence. I recommend inserting a full stop after 'body size'.
CHANGES MADE: We agree; this sentence has been split as the reviewer suggests.

30.) Line 430 - "Violins": Consider using "violin plots" instead.
CHANGES MADE: We agree; this change has been made.

31.) Fig.1 - Consider dropping the boxes around the maps and drawing them around the portions representing preservation scenarios, instead - might result in a cleaner impression.
CHANGES MADE: We agree; we have removed the boxes around maps and enlarged images so that they're easier to see and interpret.

32.) Fig. 3 - Labels on the figure mismatch the caption (I suppose this is an older version of the figure).
CHANGES MADE: The reviewer is correct – we apologize; we suspect the lead author uploaded a previous version of Figure 3 with the revised manuscript (see also our response to point 14). We have revised Figure 3 along with the caption, and hope these issues are now fixed.

33.) Fig. 4 - Note that the meaning of the word "baseline" is not the same in the caption and figure. I recommend renaming the column names from baseline and results to 'Complete preservation' and 'Experiment 5 - Casting'. Also please mention in the caption how many sites were used get the estimates.
CHANGES MADE: We agree; rather than 'complete preservation' and 'Experiment 5 – Castings', however, we have reworded these column names to "True beta" and "Simulated beta (5 – 'castings')", and updated this language throughout the manuscript. We feel this is better, and also provides the best possible match with the phrasing in the main text and captions for the other figures.

34.) Supplementary material: There is a larger chunk of text between ESM Fig. 3 and 4. Consider making a separate section from this material (e.g. supplementary text). Also, please edit the axis and panel labels for ESM 5.
CHANGES MADE: We agree; we have moved the text into a new section (ESM 5), and renumbered the other supplementary materials. We have fixed axes and panel labels for the figure shown in ESM 6.

35.) Supplied code: I deeply value that the authors shared their final code. Unfortunately, I did not have the time to go through all of it and adjust the code so it would run on my computer. The utility of this material could be enhanced by:

- structuring the files in relevant directories;
- increasing portability, i.e. decreasing the number of instances where code had to be manually edited to make it run;
- clearly indicating what the execution order of the code is to gain all results, and the purpose of individual scripts;
- and recording the versions of the used software packages and citing them as supplementary references.

CHANGES MADE: We agree, and thank the reviewer for making this suggestion. We have taken several steps to make the analyses more accessible and repeatable. First, we have organized the materials uploaded to Dryad into four distinct directories (downloadable as zip files), specifically: code, .shp files, data files, and results. Second, we have heavily annotated our codes in a fashion that should make it easy for readers to follow and run on their own. Lastly, we have provided a README.rtf file that describes the functionality of the three code files (and associated datasets), and details what readers will require to reproduce the analyses on their own. Lastly, we have provided citations and version information for all R packages used in the analyses, both in the ESM document, and in the README file uploaded to Dryad. We note that the only files we were unable to upload to Dryad are the IUCN Redlist .shp files for mammal ranges, as providing these would violate the terms of use. However, the list of mammals used in our analyses is provided in several other places, and so ranges can be downloaded individually by readers interested in reproducing our results.

We are unsure at this point if the updated files have been made available to reviewers on Dryad; in the case that they haven't, replicate folders have been uploaded to Github, and can be accessed here: <https://github.com/simondarroch/TerrestrialBiogeogSimulations>.